# Persistent β-Hexachlorocyclohexane Exposure Impacts Cellular Metabolism with a Specific Signature in Normal Human Melanocytes

**DOI:** 10.3390/cells13050374

**Published:** 2024-02-21

**Authors:** Federica Papaccio, Silvia Caputo, Alessandra Iorio, Paola De Simone, Monica Ottaviani, Antonella Del Brocco, Pasquale Frascione, Barbara Bellei

**Affiliations:** 1Cutaneous Physiopathology and Integrated Center of Metabolomics Research, San Gallicano Dermatological Institute, IRCCS, 00144 Rome, Italy; federica.papaccio@ifo.it (F.P.); silvia.caputo@ifo.it (S.C.); monica.ottaviani@ifo.it (M.O.); 2Oncologic and Preventative Dermatology, San Gallicano Dermatological Institute, IRCCS, 00144 Rome, Italy; alessandra.iorio@ifo.it (A.I.); paola.desimone@ifo.it (P.D.S.); pasquale.frascione@ifo.it (P.F.); 3Laboratory Clinimed, Clinical and Microbiological Analyses Laboratory, 03023 Ceccano, Italy; adb.clinimed@gmail.com

**Keywords:** β-hexachlorocyclohexane, melanoma, skin cancer, mitochondria, metabolism

## Abstract

Background: Cutaneous melanoma arises from skin melanocytes and has a high risk of metastatic spread. Despite better prevention, earlier detection, and the development of innovative therapies, melanoma incidence and mortality increase annually. Major clinical risk factors for melanoma include fair skin, an increased number of nevi, the presence of dysplastic nevi, and a family history of melanoma. However, several external inducers seem to be associated with melanoma susceptibility such as environmental exposure, primarily unprotected sun experience, alcohol consumption, and heavy metals. In recent years, epidemiological studies have highlighted a potential risk of β-hexachlorocyclohexane (β-HCH), the most studied organochlorine pesticide, causing cancer induction including melanoma. Methods: We evaluated in vitro the impact of this pollutant on epidermal and dermal cells, attempting to describe mechanisms that could render cutaneous cells more prone to oncogenic transformation. Results: We demonstrated that β-HCH impacts melanocyte biology with a highly cell-type specific signature that involves perturbation of AKT/mTOR and Wnt/β-catenin signaling, and AMPK activation, resulting in lowering energy reserve, cell proliferation, and pigment production. Conclusions: In conclusion, long-term exposure to persistent organic pollutants damages melanocyte metabolism in its function of melanin production with a consequent reduction of melanogenesis indicating a potential augmented skin cancer risk.

## 1. Introduction

Cutaneous melanoma is the most aggressive form of skin cancer due to its marked invasiveness which causes about 80% of the deaths resulting from this type of cancer. It develops from melanocytes which represent melanin producers. Globally, the incidence of melanoma is increasing despite conspicuous efforts in prevention strategies and population screening (secondary prevention) [1,2]. Genetic factors play an indirect role in the risk for melanoma caused by the impact on pigmentary characteristics (skin color, red hair, and freckles, number of nevi, and propensity to sunburn but not tan) [3] in addition to other heritable risk factors including a family history of melanoma [4]. Germline mutations of the cyclin-dependent kinase inhibitor 2A (*CDKN2A*) gene, which encodes two tumor suppressor proteins (p16INK4a and p14ARF) involved in cell division control, cyclin-dependent kinase-4 (*CDK4*), telomerase reverse transcriptase (*TERT*), and the protection of telomere-1 (*POT1*) genes, are associated with melanoma [5]. Among the genes that bear a moderate risk are melanocortin-1 receptor (*MC1R*) and microphthalmia-associated transcription factors (*MITF*), whose corresponding proteins are involved in the melanin biosynthetic pathway which occurs in specialized organelles named melanosomes [5]. Melanomagenesis gives one of the best examples of how genetic and environmental factors interact in the pathogenesis of cancer. Interestingly, *MC1R* polymorphic variants, particularly co-presence of multiple *MC1R* variants and red hair color variants, may increase the penetrance of *CDKN2A* mutations and the risk of cutaneous melanoma in affected families [6]. In terms of environmental factors, ultraviolet radiation (UVR) exposure is the predominant risk factor [7,8]. Individual genetic low skin pigmentation features (implying individual sun sensitivity) predict melanoma risk regardless of UVR exposure levels, evidencing that intrinsic and extrinsic factors are critically interconnected in melanoma incidence [9]. The emerging evidence suggests that oxidative stress is highly involved in melanoma formation. Skin with higher pheomelanin levels, in comparison to skin with higher eumelanin levels, tends to produce more reactive oxygen species (ROS), which can promote carcinogenesis [10]. However, exposure to other environmental factors, including particulate matter present in air pollution, pesticides, and toxins present in the atmosphere, food, and water (collectively indicated with ultraviolet radiation as skin exposome), play a relevant role in determining the overall melanoma risk [11]. Particularly, continuous exposure to organic pollution agents causes the accumulation of molecules that irreversibly compromise skin health [12]. Organochloride pesticides (OCPs), a heterogeneous class of synthetic pesticides that belong to a group of chlorinated hydrocarbon derivatives, are well-known to be 40% of total environmental pollutants, constituting a critical element for carcinogenesis [13]. Between the 1970s and 1980s, the US EPA banned or restricted their use due to concern about their remarkable environmental persistence and possible health effects [14]. However, OCP insecticides like dichloro-diphenyl-trichloroethane (DDT), hexachlorocyclohexane (HCH), aldrin, and dieldrin are still widely used in developing countries in Asia and Africa, even though, in 2009, the Stockholm convention blacklisted the OCPs as a consequence of their harmful impact on the health of humans [15,16]. OCPs are defined as persistent organic pollutants (POPs) since once released into the environment they remain for long periods with an inevitable spreading, accumulation, and biomagnification in water, soil, vegetables, farm animals, and derivatives such as milk and butter, posing a public health concern. In humans, because of the lipophilic nature of OCPs, the accumulation of these molecules has been documented in the adipose tissue in addition to plasma samples [17,18,19]. Both plasma and adipose tissue depots of OCPs are significantly associated with cancer risk [20,21]. Among the hexachlorocyclohexane isomers (α, β, γ, δ, ε) produced during the industrial synthetic process, only γ-HCH has insecticide properties and it is commonly named lindane [22]. Unlike the other HCH isomers, the specific tridimensional arrangement of the chlorine atoms on the cyclohexane ring confers on the β isoform elevated stability and the ability to accumulate in fatty tissue (10 to 30 times higher than isomer γ) which contributes to the long biological half-life and high enhancement propensity of this molecule [22]. Carcinogenic activity of β-HCH is sustained by interference at the epigenetic level of DNA function, DNA damage, perturbation of intracellular homeostatic redox equilibrium, modulation of STAT3-mediated oncogenic pathways, activation of the aryl hydrocarbon receptor (AhR), and disruption of the androgen receptor (AR) signaling cascade [20,21,22]. In addition to potential oncogenesis, constant contact with these chemical substances has been associated with a broad range of adverse effects, including reproductive defects and behavioral changes, which are believed to be related to their ability to disturb the functions of certain hormones, enzymes, mitogens, neurotransmitters, and to induce key genes involved in the metabolism of steroids and xenobiotics [23,24]. However, the available information based on in vitro studies does not fully explain epidemiological data and further studies are necessary to support public health strategies. In 2021, Darvishian and colleagues demonstrated a statistically significant increased risk of developing melanoma in patients with elevated concentrations of OCPs, and increased plasma concentrations of these pesticides in melanoma patients [25]. During the last decade, an ample epidemiological analysis conducted in a population living in the area of “Valle del Sacco” confirmed previous studies which linked the incidence of different diseases, including cancers, and the occurrence of β-HCH contamination [26,27]. Starting from these pieces of evidence and considering the alarming impact of OCP exposure on public health, in this study, we attempt to investigate the possible role of β-HCH in sustaining melanoma onset. To follow this aim, we analyzed the long-term effects of β-HCH molecules on melanocytes, fibroblasts, and keratinocytes isolated from neonatal subjects lacking significant previous conditioning by environmental factors.

## 2. Materials and Methods

### 2.1. Ethics Statement

In line with the Declaration of Helsinki Principles, patients gave written informed consent to collect samples of human material for research. Further, the Institutional Research Ethics Committee (Istituto Regina Elena e San Gallicano) approved all research activities involving human subjects.

### 2.2. Cell Cultures and Treatments

Primary cultures of normal human melanocytes (NHM), keratinocytes (NHK), and fibroblasts (NHF) were isolated from human neonatal foreskin fragments obtained during routine circumcisions. NHM were cultured in 254 Medium supplemented with HMGS (Human Melanocyte Growth Supplement) (Cascade Biologics Inc., Portland, OR, USA). NHK were maintained in M154 supplemented with HKGS (Human Keratinocyte Growth Supplement) (Cascade Biologics Inc.). NHF were cultured in DMEM (EuroClone S.p.A., Milan, Italy) supplemented with 10% fetal bovine serum (FBS) (EuroClone S.p.A.). All cell cultures were additionally supplemented with glutamine and penicillin/streptomycin (Gibco Hyclone Laboratories, South Logan, UT, USA). β-HCH (Merck, Sigma-Aldrich, Milan, Italy) was dissolved in methanol to prepare a stock solution of 10 mM and added to the cell growth medium at the final concentrations of 10 µM, 50 µM, and 100 µM. The concentration of 10 μM used for most of the experiments in this study was chosen based on previous studies [28,29] and population-based epidemiological investigations conducted by the Epidemiological Department of Lazio in the industrial area named Sacco River (Valle del Sacco; south of Rome, Italy), a well-known contaminated site [26,27]. Treatments with β-HCH were renewed every 48 h. Phenylthiourea (PTU) (Merck, Sigma-Aldrich) was dissolved in absolute ethanol to a stock solution of 50 mM and added to the cells at a final working concentration of 300 μM. The α-MSH hormone (Merck, Sigma-Aldrich) was suspended in pure H_2_O to a stock solution of 10^−7^ M and added to the cells at the final concentration of 10^−4^ M. To assess the cumulative effect of β-HCH and other additional environmental stressors, β-HCH-pretreated cells were seeded in 12-well plates at a density of about 60%. After 24 h, cells were irradiated or treated with hydrogen peroxide (H_2_O_2_) (Merck, Sigma-Aldrich). For UV irradiation and H_2_O_2_ treatment, period cells were incubated in the medium without phenol-red and FBS in the case of fibroblast. UVA was used at a dose of 6 J/cm^2^ and UVB at a dose of 40 mJ/cm^2^ using a Bio-Sun irradiation apparatus (Vilbert Lourmat, Marne-la-Vallée, France). H_2_O_2_ (Merck, Sigma-Aldrich) treatment was performed with a final concentration of 100 μM for 1 h. Preliminary experiments were performed to select a moderate cytostatic effect and to avoid cell death. Control cells were incubated in parallel without irradiation or H_2_O_2_ treatment. During the 24 h following the treatments, cells were maintained in a regular medium without any additional treatment.

### 2.3. Cell Count

An amount of 10 × 10^3^ fibroblasts, 50 × 10^3^ keratinocytes, and 40 × 10^3^ melanocytes were seeded on 12-well plates; after one day the growth medium was replaced with fresh medium containing β-HCH (or none for control cells) at the appropriate concentrations. At the endpoint (1 week), cells were detached with trypsin and washed with 1× PBS plus 1% FBS. After centrifugation, cells were resuspended in an equal volume of PBS and counted by MACSQuant Analyzer 10 Flow Cytometer (Miltenyi Biotec, S.r.l., Bologna, Italy). Each experiment was performed in duplicate.

### 2.4. MTT Assay

Samples of 10 × 10^3^ fibroblasts, 50 × 10^3^ keratinocytes, and 40 × 10^3^ melanocytes were seeded to ensure active proliferation until the endpoint was reached (1 week). Keratinocytes, fibroblasts, or melanocytes were plated into 12-well plates for 24 h to adhere. Then, the growth medium was replaced with fresh medium containing treatments (or none for control cells) at the appropriate concentrations. Culture medium and drugs were refreshed twice a week. For UV-irradiated and H_2_O_2_-treated cells, metabolic damage was evaluated after 24 h. At the experimental endpoint, cells were incubated with 3-(4,5 dimethylthiazol-2-ii)-2,5-diphenyl tetrazolium bromide (MTT) (Merck, Sigma-Aldrich) at a concentration of 5 mg/mL for 3 h. After this time, the medium was discarded and the resulting crystals were solubilized in DMSO. The absorbance was measured at 570 nm. All data was blind adjusted before statistical analysis. Each experiment was performed in duplicate.

### 2.5. Melanin Content Determination

Next, cell pellets were dissolved in 100 µL of 1 M NaOH for 2 h at 80 °C and their absorbance was measured spectrophotometrically at 405 nm using a plate reader. A standard curve using synthetic melanin (0–250 µg/mL) was used to extrapolate quantitative data. Melanin production was calculated by normalizing the total melanin values with protein content (µg melanin/mg protein).

### 2.6. Western Blot Analysis

Cell extracts were prepared with RIPA buffer containing proteases and phosphatase inhibitors. Proteins were separated on SDS-polyacrylamide gels and transferred to nitrocellulose membranes, which were saturated using EveryBlot Blocking Buffer (BioRad, Laboratories, Milan, Italy) for 10 min at room temperature and then incubated with the appropriate primary antibodies: Mitf (Dako-Agilent, Cernusco sul Naviglio, Italy), Tyrosinase (Santa Cruz Biotechnology, Inc., Dallas, TX, USA), Cofilin (BioRad, Laboratories), pAMPK, pAKT, pS6, pmTOR, LC3, Hexokinase I, Hexokinase II, PKM1,2 and PFKP (Cell Signaling Technology, Danvers, MA, USA), β-catenin (Santa Cruz Biotechnology), followed by horseradish peroxide-conjugated goat anti-mouse or goat anti-rabbit secondary antibody (Cell Signaling Technology). Imaging and densitometry analysis were performed with the UVITEC Mini HD9 acquisition system (Alliance UVItec Ltd., Cambridge, UK).

### 2.7. Gene Expression Analysis

Total RNA was extracted using Aurum Total mini kit (BioRad, Laboratories) and cDNA was synthesized using the PrimeScriptTM RT Master Mix (Takara Bio Inc., Nojihigashi, Japan). For semi-quantitative real-time PCR, cDNA was amplified using SYBR qPCR Master Mix (Vazyme Biotech Co., Ltd., Nanjing, China) containing 25 pmol of forward and reverse primers. Reactions were carried out using a CFX96 Real-Time System (Bio-Rad Laboratories). All samples were tested in triplicate. Amplification of the β-actin transcript was included in all samples as an internal control. Sequences of primers used were the following: β-actin forward, 5′-GACAGGATGCAGAAGGAGATTACT-3′; reverse 5′-TGATCCACATCTGCTGGAAGGT-3′; Mitf forward, 5′-ATGGACGACACCCTTTCTC-3′; reverse 5′-GGAGGATTCGCTAACAAGTG-3′; Tyrosinase forward, 5′-GGCCAGCTTTCAGGCAGAGGT-3′; reverse 5′-TGGTGCTTCATGGGCAAAATC-3′. 

### 2.8. Flow Cytometry Analysis

ROS detection: The relative level of intracellular ROS was measured with the redox-sensitive fluorescent dye 2′7′-dichlorodihydrofluorescein diacetate (H_2_DCFDA; Sigma-Aldrich). Cell-permeable, non-fluorescent H_2_DCF is rapidly oxidized to highly fluorescent dye 2′7′-dichlorofluorescein (DCF) in the presence of intracellular ROS. Cells were incubated with 2.5 μM H2DCF for 30 min at 37 °C and 5% CO_2_ in phenol red-free full-starved medium in the dark. After removing the probe solution, cells were washed with PBS, trypsinized, centrifuged at 800 rpm, and then suspended in PBS. The intensity of the fluorescent signal was detected by the MACSQuant Analyzer 10 Flow Cytometer (Miltenyi Biotec) in the FL1 channel. The data, reported as mean fluorescence intensity in histogram plots ± SD, are presented as a percentage of control. Unstained cells were used as a negative control. Data were collected from at least three independent experiments.

Assessment of mitochondrial membrane potential (∆Ψ): Cells were incubated with 2 μM of dye JC1 (5′,6,6′-tetrachloro-1,1′,3,3′-tetraethylbenzimidazolylcarbocyanine iodide) (ThermoFisher Scientific, Waltham, MA, USA) for 30 min at 37 °C and 5% CO_2_ in phenol red-free full-starved medium in the dark. After removing the probe solution, cells were washed with PBS, trypsinized, centrifuged at 800 rpm, and then suspended in PBS. Double fluorescence generated by JC1 staining of mitochondria, corresponding to the green fluorescent J-monomers and the red fluorescent J-aggregates, was used for monitoring the mitochondrial membrane potential. Signals were measured by MACSQuant Analyzer 10 Flow Cytometer (Miltenyi Biotec) in channels FL and FL2 respectively. Results are reported as FL2-PE/FL1-FITC fluorescence intensity ratio ± SD. Unstained cells were used as a negative control. Data were collected from at least three independent experiments.

Mitochondrial Mass measurement: Cells were incubated with 0.1 μM of MitoTracker^®^ dye (ThermoFisher Scientific) for 30 min at 37 °C and 5% CO_2_ in phenol red-free full-starved medium in the dark. After removing the probe solution, cells were washed with PBS, trypsinized, centrifuged at 800 rpm, and then suspended in PBS. Fluorescence signals were measured by MACSQuant Analyzer 10 Flow Cytometer (Miltenyi Biotec) in channel FL1 and reported as mean fluorescence intensity ± SD in histogram plots. Unstained cells were used as a negative control. Data were collected from at least three independent experiments.

### 2.9. Annexin V/PI Staining

Cell death and apoptosis were evaluated by the annexin-V FITC/propidium iodide (PI) double staining method after 1 week of treatment. At the end point, cells were harvested by trypsinization, suspended in the staining buffer (10 mM HEPES/NaOH, pH 7.4, 140 mM NaCl, 2.5 mM CaCl_2_), stained with FITC-labeled Annexin V I for 15 min at room temperature in the dark. Samples were then washed with PBS, centrifuged, suspended in PBS containing PI, and then kept on ice until the analysis by MACSQuant Analyzer 10 Flow Cytometer (Miltenyi Biotec), channels FL1 and FL3. Unstained cells were used as a negative control. The percentages of single-positive (Annexin V, early apoptosis) and double-positive (Annexin V plus PI, late apoptosis) cells that were added are reported in Table 1.

### 2.10. ATP Determination

The intracellular level of ATP was measured using a commercial fluorometric kit (ThermoFisher Scientific, Monza, Italy) according to the manufacturer’s instructions. At each experimental endpoint, considered untreated control cells were analyzed. The results obtained as μM after normalization for protein concentration were reported as fold-change over control mean ± SD.

### 2.11. MC1R Genomic Characterization

DNA sequencing was used for screening polymorphic variants of *MC1R* (NM_0002386). Genomic DNA was extracted from melanocytes using a Tissue Kit (Qiagen, Milan, Italy) following the manufacturer’s instructions. About 100–200 ng of genomic DNA was amplified by PCR in a total volume of 50 μL containing 25 μL of 2× AmpliTaq GoldTM 360 Master Mix (Thermo Fisher Scientific) and 25 pmol of each primer. Primer sets were designed to cover the entire coding sequences plus a few nucleotides into sequences on both ends. PCR primer sequences are available upon request. DNA fragments were checked by electrophoresis in 2% agarose gel and purified before bidirectional direct Sanger sequencing and chromatogram inspection using ChromasPro software version 2.6.6.

### 2.12. Extraction and GC-MS Analysis of β-HCH

To assess β-HCH uptake, cells were detached, counted, washed twice with PBS, collected, and stored at −80 °C until analysis. The removal of β-HCH from melanocyte and fibroblast pellets was achieved with chloroform/methanol (2:1) after suspending pellets in distilled water. The liquid/liquid extraction was performed twice and the collected organic layers were evaporated under nitrogen flow. The dried extracts were dissolved in methanol and analyzed by gas chromatography-mass spectrometry (GC-MS) to establish the β-HCH amount in the cells (GC 7890A coupled to MS 5975 VL analyzer, Agilent Technologies, Santa Clara, CA, USA). Chromatographic separation was performed on an HP-5MS (Agilent Technologies) capillary column (30 m × 250 µm × 0.25 µm), using helium as the carrier gas (1 mL/min). The GC oven temperature program was as follows: 10 °C for 1 min; 100–180 °C at 10 °C/min; 180–225 °C (2 min) at 2 °C/min; 225–265 °C (1 min) at 20 °C/min. The temperature of the injector, ionization source, quadrupole, and transfer line were kept at 250, 230, 150, and 300 °C, respectively. Samples were analyzed in scan mode, utilizing electron impact (EI) mass spectrometry. The comparison with the β-HCH authentic standard and the match with the spectral library allowed the identification of β-HCH in the samples. Total ion chromatograms (TIC) were acquired, and areas of single peaks, corresponding to β-HCH, were integrated with the qualitative analysis software. An external standard calibration curve, constructed using untreated cells as the matrix, was used for the quantitative analysis. The results were expressed as nmol/10^6^ cells.

### 2.13. Statistical Analysis

Quantitative data were reported as mean standard deviation (SD). Student *t*-test was used to assess statistical significance with thresholds of * *p* ≤ 0.05 and ** *p* ≤ 0.01.

## 3. Results

### 3.1. β-HCH Impacts the Proliferation Capacity of Skin Cells

Based on previous data [29], we started our study by testing the effect of β-HCH on the proliferation of cutaneous cells. Further, a possible cytotoxic effect was evaluated by MTT assay, since it measures the mitochondrial respiration dependent on the catalytic activity of mitochondrial succinate dehydrogenase enzyme. Although the MTT assay results generally correlated with the number of viable cells growing in standard culture conditions, the rate of tetrazolium reduction reflects precisely the general metabolic activity and the rate of glycolytic NADH production and might reveal subtoxic energetic disequilibrium. Skin melanocytes, fibroblasts, and keratinocytes were treated with β-HCH for 1 week within a range of concentrations from 10 μM to 100 μM. The dosage of 10 μM corresponds to the plasmatic level of β-HCH measured in patients participating in the epidemiological study conducted in the “Valle del Sacco” [27] and, representing a realistic model, has been extensively used in previous studies [28,29,30,31]. Here, after 1 week of continuous exposure, cell count showed an augmented proliferation rate in keratinocytes and fibroblasts treated with 10 and 50 μM, whereas the dose of 100 μM further accelerated cell division in keratinocytes but exerted an opposite effect on fibroblasts (Figure 1a,b). In contrast, β-HCH significantly reduced melanocyte proliferation at all the concentrations tested in a dose-dependent manner (Figure 1c). Microscopic examination of cell cultures before performing cell enumeration assay did not reveal detached or dead cells (At the same time, annexin/PI staining confirmed the modest number of apoptotic events in the presence of β-HCH compared to untreated melanocytes (Figure 1d), therefore we concluded that this type of cell mostly sustained a cytostatic effect. Similar results were obtained after 2 and 3 weeks. MTT assay is an indicator of cell number and viability; however, since it depends on metabolic rate and number of mitochondria in addition to cell number, it does not necessarily correlate to cell count [32]. In our experiments, β-HCH lowered intracellular MTT conversion into formazan at all the concentrations used and in all the cell types tested (Figure 1e–g), indicating that only in melanocytes was the overall absorbance measured (OD) with MTT assay proportional to the number of viable cells. In contrast, keratinocyte and fibroblast cell cultures evidenced a lack of linear direct proportion of MTT results and the number of cells (Figure 1h–j), suggesting the activation of rescue pathways that result in stimulation of cell proliferation. 

Consequently, the ratio of metabolic activity/cell number assumed a value <1, proposing a cytostatic energy-saving mechanism in keratinocytes and fibroblasts at low doses with a paradoxical positive value in cells suffering β-HCH toxicity (melanocytes at all the concentrations and fibroblasts exposed to the higher dose) (Table 1).

Taken together, these data lead to the belief that β-HCH primarily impacts intracellular metabolic activity leading to metabolic stress which, if not adequately compensated (such as in the case of pigment cells), imposes growth arrest.

### 3.2. β-HCH Selectively Affects Melanocyte Intracellular Metabolism

To characterize the impact of β-HCH on mitochondrial activity, we measured the amount of ATP, since changes in ATP levels could reflect defects in mitochondrial energy production. For this purpose, intracellular ATP variations were monitored weekly during a period of 6 weeks of β-HCH treatment at the dose of 10 µM. In melanocyte cultures, we detected a gradual time and dose-dependent reduction of ATP content that became significant starting from the second week (Figure 2a,b).

By contrast, keratinocytes and fibroblasts treated for two weeks with 10 µM β-HCH maintained an overall unmodified level of ATP (Figure 2c), suggesting a specific melanocyte-associated metabolic disequilibrium. To confirm this hypothesis and to exclude a different kinetic, fibroblast treatment was protracted until 12 weeks (Figure 2d). Given this evidence, we also performed dosage of ATP in melanocytes managed for a long period with a common inhibitor of tyrosinase enzyme, phenylthiourea (PTU). The darker melanocyte cell line, NHMp111, was treated for three weeks with 300 μM of PTU to abrogate melanin production and obtain a significant reduction of intracellular pigment accumulation (Figure 2e). Subsequently, these cells were then exposed to β-HCH for two weeks until ATP evaluation. The lack of ATP decrease in depigmented melanocytes additionally linked the β-HCH-dependent toxicity to the melanogenic differentiation (Figure 2f). Moreover, according to the idea that the melanogenic biosynthetic pathway is a strongly energy-consuming process, data confirmed that when melanogenesis is repressed, the intracellular energetic reserve is superior (Figure 2f). To further analyze the relationship between β-HCH-associated damage and melanocyte lineage-specific features, all five melanocyte cultures used were characterized for melanocortin-1 receptor (*MC1R*) genotype and melanin content. *MC1R* is the key regulator of the synthesis of epidermal melanin pigment, and the polymorphic isoform significantly contributes to physiological and pathological variation of pigmentation and sensitivity to UV-dependent damage [33]. Among the melanocyte cultures included in the study, we found three natural single nucleotide *MC1R* allelic variants: Asp84Glu (NHMp1), Val92Met (NHMp13), Val60Leu (NHMp70), and two wild-type gene sequences (NHMp76 and NHMp111). Consistent with genetic data, melanocytes carrying *MC1R* variants and exhibiting consequent dysfunctional *MC1R* intracellular signaling displayed a general reduction of melanin compared to melanocytes with wild-type *MC1R* gene (Figure 2g). Of note, NHFp70 carrying Val60Leu variant presented an unexpected intracellular melanin load similar to cells with wild-type genotype. In apparent contradiction to the idea that melanogenesis is responsible for β-HCH-associated metabolic damage, data obtained by clustering cells based on melanin content showed that in poorly and moderately pigmented melanocytes (NHMp1, NHMp13, NHMp76) the decrease of ATP is an early event, whereas highly pigmented cells (NHMp70 and NHMp111) were initially more resistant to the impairment in ATP content (Figure 2h). Treatments with the higher β-HCH dose (50 μM) confirmed dose, time, and pigment-dependent toxicity (Figure 2i). However, the quantification of intracellular ROS, another marker of mitochondrial metabolic stress, demonstrated a general perturbation of redox equilibrium (Figure 3a).

Furthermore, all cell cultures showed a hyperpolarization of the mitochondrial transmembrane potential as recorded by JC1 staining (Figure 3b) and an augmented mitochondrial mass measured with Mitotracker Green staining (Figure 3c). Interestingly, a significant persistent mitochondrial membrane hyperpolarization, a condition that further exacerbates oxidative stress, has been previously associated with the defective functionality of ATP synthase and ATP depletion [34,35]. By contrast, in β-HCH-treated keratinocytes and fibroblasts, the amount of ROS, the mitochondrial mass, and the mitochondrial membrane potential were not perturbed by β-HCH treatment (Figure 3d–f). To confirm the specific signature of melanocytes and exclude differences related to the genetic background, fibroblasts isolated from the same subjects of darker melanocyte cell cultures were tested, confirming no variation in ROS level, Mitotracker Green, and JC1 staining (Figure 3g–i). In melanocytes, after two weeks of β-HCH exposure the contraction of anabolic activities and the concomitant compensatory augmented catabolic metabolism was evidenced by the mechanistic target of rapamycin (mTOR) inhibition, which corresponded to a mild decrease in phosphorylation of its main target, the p70 ribosomal S6 kinase of pAKT, and to a higher amount of the faster electrophoretic mobility isoform of LC3 (LC3-II), a well-established marker for phagophores and autophagosomes (Figure 4a). Of interest, both the phosphorylated active form and the total amount of mTOR were reduced by β-HCH. In line with chronically compromised mitochondrial activity, the main sensor of cellular energy state, AMPK (AMP-activated protein kinase), which is capable of promoting adaptive responses during critical stress periods, became hypophosphorylated. Thus, in some way, β-HCH inhibits the allosteric activation of AMPK by AMP and ADP. Also, a light but constant augmentation of phosphorylation activation of ERK1/2 was observed, suggesting the activation of MAPKs signaling. Further investigation of glycolytic enzyme expression evidenced reduced expression of Hexokinases I, II, of Pyruvate kinase (PKM1/2), and of Platelet-type phosphofructokinase (PFKP) (Figure 4b). Of interest, Hexos are associated with the outer mitochondrial membrane and are critical for maintaining an adequate rate of glycolysis. By contrast, Glyceraldehyde-3-phosphate dehydrogenase (GAPDH) and pyruvate dehydrogenase (PD) were unaffected by the treatments.

The significance of autophagy activation might reside in the necessity to eliminate damaged macromolecules and organelles, as well as in providing energy in response to metabolic and environmental stress [36]. The evidence that AMPK, the protein responsible for sensing metabolic stress (particularly falling cellular energy status signaled by rising AMP/ATP and ADP/ATP ratios) was not activated in β-HCH-treated cells, suggesting a lack of coordination between catabolic and anabolic processes. In fact, eukaryotic cells have opposing roles in cell proliferation, metabolic regulation, and autophagy signaling pathways; AMPK and mTOR are physiologically regulated antithetically [37,38]. 

### 3.3. Chronic β-HCH Exposition Reduces Melanin Synthesis

Melanogenesis is a complex process comprising many energy-consuming steps. Thus, we addressed the question of whether, in the case of β-HCH-dependent reduction of ATP, melanocytes compensate for the global cellular energetic request by reducing melanin synthesis. As shown in Figure 5a, independently from the initial amount of melanin, darkly and lightly pigmented melanocytes proportionally decreased the amount of intracellular melanin. 

However, the expression of melanocyte-inducing transcription factor (*MITF*), the master regulator of melanocyte differentiation, and tyrosinase, the enzyme responsible for the initial step of melanin synthesis, were increased in the presence of β-HCH at the mRNA and protein level (Figure 5b,c), suggesting that other mechanisms are responsible for lower melanin amounts. It is possible that in the context of suboptimal mitochondrial functionality, to efficiently supply ATP into sites of critical high-energy demand processes such as those occurring for survival, melanocytes adapt their metabolism by finely modulating melanogenic signaling. In line with the idea that melanogenesis implies metabolic remodeling, stimulation of melanocytes carrying a wild-type *MC1R* gene with its natural ligand, the alpha-melanocyte stimulating hormone (α-MSH), in starved medium activates mTOR by phosphorylation and its downstream target pS6 (Figure 5d). Notably, hormonal stimulation also increased the total amount of mTOR leading to an overall unchanged pmTOR/mTOR ratio. At the same time, as expected, the engagement of *MC1R*-activated pro-melanogenic signaling includes the Wnt/β-catenin pathway and downstream tyrosinase expression (Figure 5d). All these events except the phosphorylation of S6 were scaled down in the presence of β-HCH, confirming the inadequate capacity of the intracellular metabolic milieu to support an energy-consuming process such as pigmentation. Further, in a starved medium, β-HCH exerted opposite effects on AMPK depending on the presence of α-MSH. Here, in the absence of growth factors (low energetic demand), the presence of β-HCH sustained AMPK activation whereas, after stimulation with α-MSH (increased metabolic demand), β-HCH down-modulated phosphorylation activation of AMPK. Again, results demonstrated that the toxicity due to β-HCH accumulation drove a protective decrease of intracellular metabolism. Since activation of Wnt/β-catenin and Akt/mTOR signaling is deeply implicated in *MC1R*-mediated survival against oxidative stress [39,40], and treatment with β-catenin agonists attenuates oxidative stress and lessens H_2_O_2_-induced cell apoptosis, we exposed melanocytes to H_2_O_2_, UVA, and UVB after 3 weeks of conditioning with β-HCH. Preliminary experiments were performed to select subtoxic doses that do not compromise cell viability. In non-pigmented cells such as fibroblasts, pretreatment with β-HCH lightly exacerbated the cytostatic effect of other environmental stimuli measured 24 h after UV and H_2_O_2_ administration (Figure 6a). 

On the other hand, in the case of melanocytes, no differences emerged when comparing β-HCH pre-treated melanocytes and control cells after UV or H_2_O_2_ exposure (Figure 6b). However, at the molecular level, oxidative damage due to UVA and β-HCH both down-modulated β-catenin expression and the combination of these factors exerted an additive effect (Figure 6c). Moreover, UVA activated both AMPK and S6 by hyperphosphorylation in β-HCH-pretreated melanocytes suggesting the occurrence of a robust metabolic impairment. At the same time, β-HCH counteracted the stimulation of AKT signaling. 

### 3.4. Melanocyte-Specific Features Impact β-HCH Intracellular Accumulation

Several reports document β-HCH accretion in different tissues including the skin. Due to the correlation between blood and skin quantity of β-HCH residues, monitoring the levels of organochlorine insecticides in the skin has been proposed as a non-invasive method to estimate β-HCH body burden [41]. Further, a realistic concentration level of β-HCH in different cell types remains completely unexplored. Here, we addressed the question of whether the presence of melanosomes loaded with melanin pigment could impact the quantity of pesticide trapped in the cytoplasm and consequently the deleterious effect of this pesticide. For this purpose, we extracted β-HCH from pigmented and from fibroblast cell cultures lacking melanosomes after 2 and 4 weeks of treatment. Data shown in Figure 7a demonstrated that in the case of shorter treatment, β-HCH accumulated into the cells reaching nanomolar concentrations/10^6^ cells independently of the cell type; however, during this time highly pigmented melanocytes lost most of the pesticides. Detailed time-dependent kinetic of intracellular accumulation of β-HCH revealed that its accretion as well its reduction proceeded faster in darker cells (Figure 7b).

## 4. Discussion

Persistent organic pollutants that may accumulate in the environment and food chain represent a serious human health risk. In the last decade, epidemiological and in vitro studies converged in indicating a potential cancer risk. Environmental factors play a relevant role in the progression of the melanocyte to malignant melanoma, a process that involves various sequential steps. Engagement of hydrocarbon receptor (AhR), oxidative stress, chronic inflammation, and DNA damage have been indicated as processes by which pesticides may augment oncological risk. However, a definite mechanism has not yet been demonstrated. Our study showed that at a concentration frequently observed in human serum and tissue β-HCH slows melanocyte proliferation, and reduces melanogenesis as part of a more general reorganization of metabolic activities. Remodeling of bioenergetic equilibrium appeared to be a consequence of the oxidative stress generated by prolonged exposure to β-HCH. Depending on the level of global pigmentation, β-HCH impacted melanocyte biology with different kinetics and intensities. At the beginning of the treatment, darker pigmented melanocytes offer major resistance to β-HCH-dependent mitochondrial damage, as demonstrated by the earlier ATP reduction in lightly pigmented cells. A possible explanation might reside in the type of melanin produced, since these *MC1R* allelic polymorphisms are associated with a higher pheomelanin/eumelanin ratio and a pheomelanin-rich phenotype is more prone to ROS formation even in the absence of UV radiation [42,43]. However, prolonged treatment significantly affected the metabolic equilibrium of all melanocyte cell cultures. The interference of β-HCH with mitochondrial function, which is postulated here, was similarly supported by a recent report [44]. Yang and colleagues demonstrated that β-HCH elevates ROS level in human sperm, leading to mitochondrial depolarization and thereby disrupting ATP production [44]. At the molecular level, we demonstrated that stimulation of the melanin biosynthetic pathway exacerbated metabolic stress. Inactivation of mTOR ensures that cells do not continue to proliferate under unfavorable conditions [45]. Moreover, shutting down mTOR signaling likely serves to save valuable energy for the most essential cell functions. Decreased mTOR activity physiologically upregulates catabolic processes to remove dysfunctional cellular components via autophagy [46]. Autophagy can regulate skin pigmentation through melanosome degradation in both keratinocytes and melanocytes [47]. Hence, augmented dysfunctional autophagic flux and consequent high melanosome degradation might explain the small amount of intracellular melanin even in the presence of the correct expression of the necessary enzymatic apparatus. At the same time, a drop of intracellular β-HCH might be the result of this exacerbated recycling of cellular elements. Moreover, the simultaneous lessening of melanin and β-HCH, as well as the unaffected accumulation of β-HCH in lightly pigmented melanocytes and non-pigmented cells, suggests a possible preferential localization of this organochloride in melanin-loaded melanosomes. Adsorption of HCH isomers has been extensively demonstrated in human serum, adipose tissue, brain, kidney, muscle, sebum, and placenta, following inhalation, oral and dermal exposure [48,49,50]. In the general population as well as in occupational workers, the accumulation of β-HCH assumed a linear increase over the time of exposure [51]. In rats, the distribution pattern for β-HCH was found to decrease in the sequence: fat > kidney > lungs > liver > muscle > heart > spleen > brain > blood [52]. Animal-based studies using dermal application mainly focused on clarifying the final destination of β-HCH isomers in the body, and little is known about skin toxicity. Being highly soluble in lipids, the amount of organochlorine pesticide in the skin has been shown to correlate with skin lipids [53]. Furthermore, as a result of its persistence in the body, the amount of skin β-HCH directly correlated with age, being higher in older subjects [53]. Due to the continual and rapid replacement of epidermal keratinocytes, the potential long-term effect of β-HCH accumulation might be a minor concern. However, β-HCH lowers the production and consequent distribution of pigment to surrounding keratinocytes which affects the skin’s capacity to counteract UV damage. Thus, in line with the principle, long-term exposure to organochlorine pesticides reducing melanin protection and promoting keratinocyte proliferation might impact non-melanoma skin cancer risk. Here, using an in vitro model simulating chronic exposure to β-HCH, we observed a rapid decline in intracellular storage of this molecule underlying the toxicity of this compound for melanocyte biology. On the other hand, fibroblasts and lightly pigmented melanocytes endured elevated β-HCH amounts, reinforcing the idea that melanosomes are a crucial cellular compartment where organochloride pesticides disrupt metabolic equilibrium. Thus, since the dark phenotype largely depends on the capacity to respond to stimulation of the pro-melanogenic peptide α-MSH and consequently on the *MC1R* sequence, an unexpectedly increased melanoma risk could be hypothesized for individuals bearing the wild-type form of this receptor. This phototype-dependent disparity points to the question of a new genome-environment interaction. Even though our study, which testing either a single dose of subtoxic UVA or UVB, does not support the idea that β-HCH exacerbates UV damage in pigmented cells, additional consideration is needed regarding the importance of photoprotection in darker skin which is naturally better protected against UVB and more prone to pigmentation induced by visible light and UVA. However, mutational events might help melanocytes to overcome the down-regulation of cell cycle progression, allowing stressed melanocytes to survive accumulating detrimental damage. Further studies are needed to investigate the combination of organochloride pesticides and repeated UV exposure as it is commonly experienced by farm workers. 

## 5. Conclusions

In conclusion, we demonstrated that β-HCH impacts melanocyte biology with a highly cell-type specific signature that involves perturbation of AKT/mTOR and Wnt/β-catenin signaling, and AMPK activation, resulting in a lowering of energy reserves, cell proliferation, and pigment production. Overall, melanocytes demonstrated a major β-HCH-derived toxicity compared to keratinocytes and fibroblasts. However, impacting the physiological protective role of melanin, melanocyte damage might spread throughout the entire skin homeostasis.

## Figures and Tables

**Figure 1 cells-13-00374-f001:**
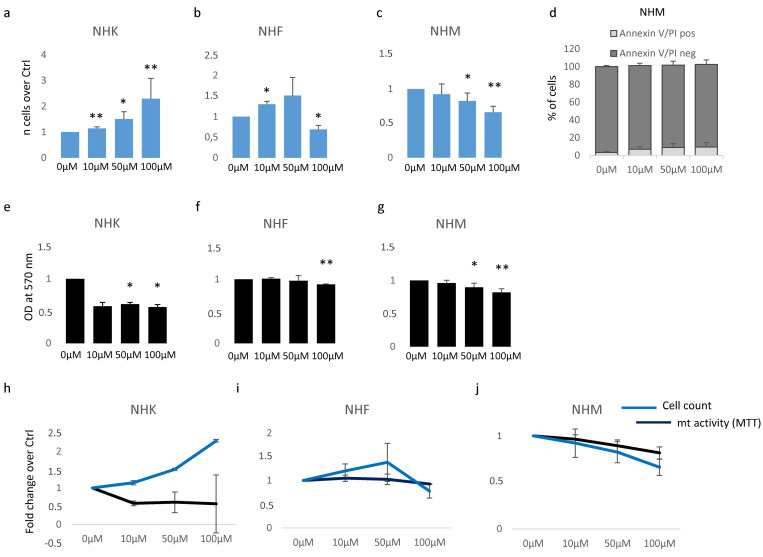
Analysis of β-HCH cytotoxicity. The impact of different doses of β-HCH on cell proliferation was evaluated following 1 week of continuous treatment with 10 μM of pesticide in NHK (**a**), NHF (**b**), and NHM (**c**). At the same end-point, Annexin V/PI staining was used to exclude apoptotic melanocytes (**d**). In parallel to cell count, MTT assay measured metabolic activity in NHK (**e**) and NHF (**f**,**g**). For comparison, cell count and MTT results were plotted together: NHK (**h**), NHF (**i**), and NHM (**j**). * *p* ≤ 0.05 and ** *p* ≤ 0.01 vs. untreated cells.

**Figure 2 cells-13-00374-f002:**
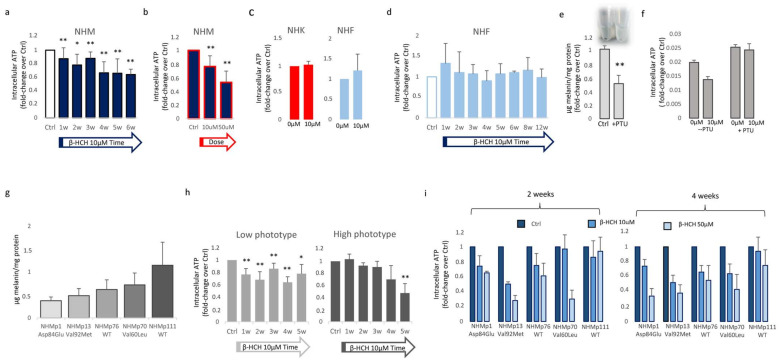
β-HCH impact the intracellular metabolic activity of melanocytes. ATP production time-dependently declined in melanocyte cultures chronically exposed to β-HCH (**a**). The assessment of ATP amounts after 2 weeks treatment with β-HCH demonstrated also a dose-dependent lowering of intracellular ATP in melanocytes (**b**). The ATP level of keratinocyte and fibroblast cultures was not affected by β-HCH after 2 weeks (**c**). Longer periods of treatment confirmed no modification of intracellular ATP in fibroblast until 12 weeks (**d**). The relative amount of total melanin was significantly reduced by long-term treatment (3 weeks) with PTU (**e**). Lightly pigmented melanocytes are resistant to ATP decline (**f**). Absolute value for intracellular melanin in five different melanocyte cell cultures carrying different *MC1R* genotypes (**g**). ATP level measured after clustering melanocyte cultures based on the melanin content (**h**). Histograms represent ATP measurements in different melanocyte cell lines after 2 and 4 weeks (**i**). * *p* ≤ 0.05 and ** *p* ≤ 0.01 vs. untreated cells.

**Figure 3 cells-13-00374-f003:**
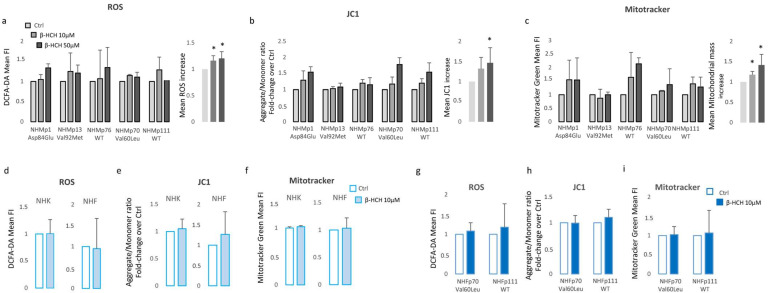
β-HCH selectively compromises oxidative equilibrium in NHM. Increase of intracellular ROS measured with DCFA-DA (**a**), hyperpolarization of mitochondrial membrane potential calculated as JC1-aggregate/JC1-monomer ratio (**b**), and augmented mitochondrial mass estimated by staining with Mitotracker Green (**c**), evidenced an overall oxidative equilibrium perturbation. For statistical analysis, data from all cell cultures were also combined in a histogram. Similarly, ROS (**d**), mitochondrial membrane potential (**e**), and mitochondrial mass (**f**) failed in the identification of any significant variation of these parameters in keratinocytes and fibroblasts. Fibroblast cell cultures isolated from the same biopsies of darker melanocytes (NHMp70 and NHMp111) confirmed the resistance of fibroblast to β-HCH-dependent metabolic impairment (**g**–**i**). * *p* ≤ 0.05 vs. untreated cells.

**Figure 4 cells-13-00374-f004:**
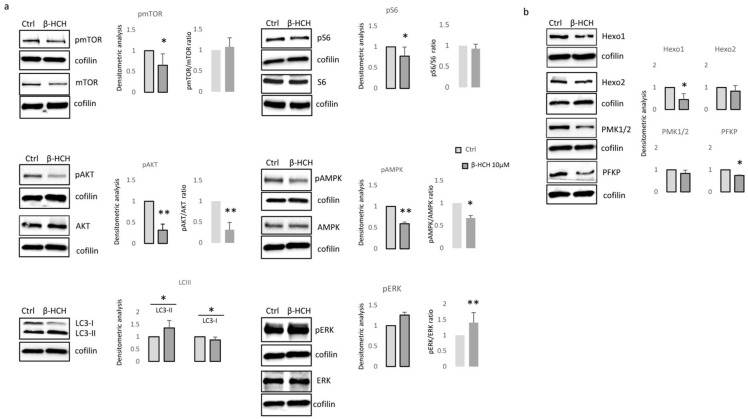
Western blot analysis of key signaling pathways involved in metabolic processes. (**a**,**b**) The darker melanocyte cell lines (NHMp70 and NHMp111) were analyzed after 2 weeks of continuous treatment with β-HCH (10 μM). Images are representative of one experiment whereas densitometric analysis, as reported in the histograms, was done comparing at least three independent experiments. * *p* ≤ 0.05 and ** *p* ≤ 0.01 vs. untreated cells.

**Figure 5 cells-13-00374-f005:**
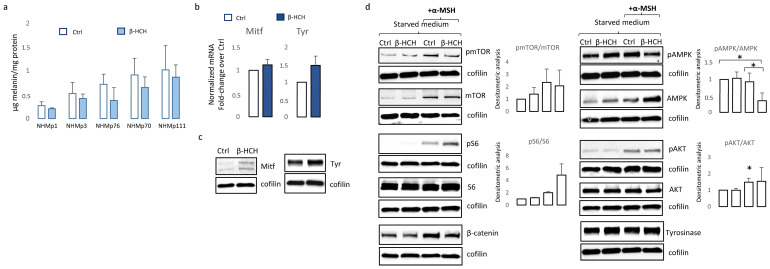
Analysis of β-HCH-dependent effect on differentiation and survival signaling pathways in melanocytes. Total melanin content was measured after 2 weeks of continuous exposure of melanocytes. Each cell line was analyzed twice (**a**). Histograms report the mean ± SD level of *MITF* and tyrosinase mRNA level of expression of all five melanocytes cell lines after 2 week of β-HCH treatment (**b**). One representative Western blot confirming the augmented expression of MITF and tyrosinase expression (**c**). Western blot analysis of NHMp111 pre-treated (or not) with 10 μM of β-HCH for 3 weeks and then treated with α-MSH 1 × 10^−4^ M for 24 h (**d**). * *p* ≤ 0.05 vs. untreated cells.

**Figure 6 cells-13-00374-f006:**
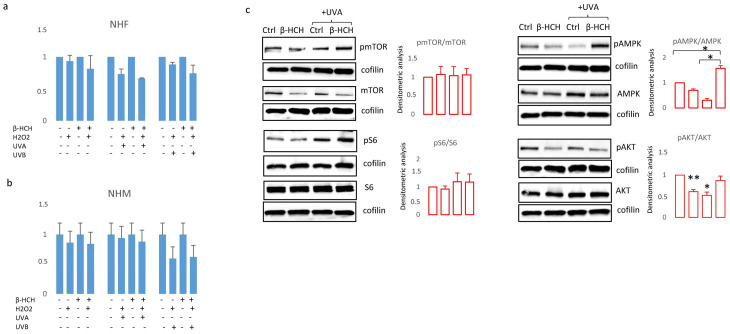
The cumulative effect of β-HCH (3 weeks exposure) on UVA (6 J/cm^2^), UVB (40 mJ/cm^2^) and H_2_O_2_ (100 μM) was evaluated using MTT assay after 24 h in NHF (**a**) and NHM (**b**). Western blot analysis of proteins implicated in metabolic and proliferative processes extracted in darker melanocyte cells 24 h after UVA exposure (**c**). * *p* ≤ 0.05 and ** *p* ≤ 0.01 vs. untreated cells.

**Figure 7 cells-13-00374-f007:**
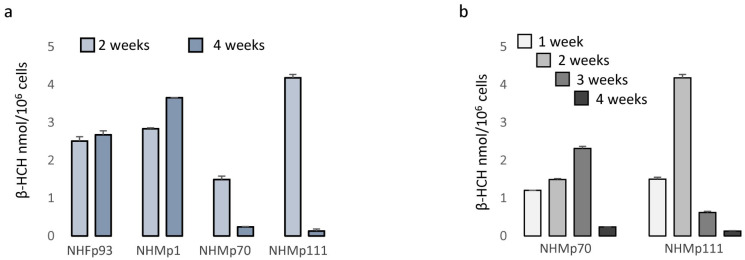
Analysis of β-HCH intracellular accumulation. The intracellular level of β-HCH in fibroblasts and melanocytes with different levels of pigmentation was measured by gas chromatography-mass spectrometry (GC-MS) and normalized for number of cells after 2 and 4 weeks of treatment (**a**). Further, a more detailed kinetic of β-HCH uptake and retention was performed on darker melanocytes (NHMp70 and NHMp111) (**b**).

**Table 1 cells-13-00374-t001:** Metabolic activity normalized for number of cells.

**β-HCH**	**10 μM**	**50 μM**	**100 μM**
NHF	0.65 ± 0.02	0.73 ± 0.26	1.11 ± 0.19
NHK	0.46 ± 0.11	0.38 ± 0.14	0.27 ± 0.15
NHM	1.05 ± 0.17	1.11 ± 0.38	1.21 ± 0.3

## Data Availability

The datasets used or analyzed during the current study are available from the corresponding author upon reasonable request.

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
