# Peer review of "Persistent β-Hexachlorocyclohexane Exposure Impacts Cellular Metabolism with a Specific Signature in Normal Human Melanocytes"

_cells, 2024, doi:10.3390/cells13050374_

Round 1

Reviewer 1 Report

Comments and Suggestions for Authors

- Although the "science" is "solid", the manuscript needs to be rewritten to improve it and not distract the reader from the background, results, and discussion. Here are some of the minor comment: There are a lot of words used that are quite "unusual", therefore making it difficult to understand; Sentences were started with a number; No spaces between values and units, etc. 

- The Figures need improvement to make them acceptable for publication- the Y- and x-axis labels and titles, the size of the figures, etc.

Comments on the Quality of English Language

Refer to the above comments

Author Response

Cells

Editorial Office

                                                                          Rome, January 26th, 2024

Revised manuscript cells-2813929

Dear Editor,

please find enclosed the revised version of the manuscript (cells-2813929-R1) entitled “Persistent β-Hexachlorocyclohexane Exposure Impacts Cellular
Metabolism With a Specific Signature in the Human Normal Melanocytes:
Implication in Melanomagenesis” by F. Papaccio et al. Firstly, we would like to thank you and the reviewers for the positive and helpful comments about our paper. We have revised the manuscript accordingly to the requests and answered the comments point-by-point as follows:

Reviewers' comments:

Reviewer 1

- Although the "science" is "solid", the manuscript needs to be rewritten to improve it and not distract the reader from the background, results, and discussion. Here are some of the minor comment: There are a lot of words used that are quite "unusual", therefore making it difficult to understand; Sentences were started with a number; No spaces between values and units, etc. 

- As requested we improved the manuscript. The spaces have been added, however we did not find number at the beginning of sentences.

- The Figures need improvement to make them acceptable for publication- the Y- and x-axis labels and titles, the size of the figures, etc.

We thank the Reviewer for his opinion. Accordingly, also the figures have been improved. Particularly, considering the quantity of data contained, Fig. 5 has been divided in two figures. Some additional informations have been added to help the readers in all the other images (except Fig.7).

Reviewer 2 Report

Comments and Suggestions for Authors

The authors studied the effect of the organochlorine pesticide β-hexachlorocyclohexane (β-HCH) on melanocytes, skin fibroblasts and keratinocytes in vitro. They showed that β-HCH increased keratinocytes and fibroblasts' proliferation whereas it inhibited melanocytes' proliferation. They further demonstrated that β-HCH selectively affected melanocyte intracellular metabolism, reducing ATP and inducing a general perturbation of redox equilibrium with a consequent reduction of melanogenesis. This was associated with a decrease in phosphorylation of several proteins involved in key signaling pathways. They concluded that exposure to β-HCH impaired melanocyte metabolism and could increase skin cancer risk.

The results are interesting to better understand the long-term effects of pesticides on skin cells. However, despite what is claimed in the title and the abstract, there is no evidence for an association of the impaired melanocyte metabolism induced by β-HCH and the increased in skin cancer risk. The authors clearly show that β-HCH inhibits melanocyte proliferation and inhibits the activation of several oncogenic pathways, which is not what you expect from an increased risk of melanocyte transformation on the contrary. From the data presented, in particular the reduction in melanin, one would expect β-HCH to induce depigmentation in vivo such as vitiligo. Is there any epidemiological evidence for increased vitiligo in population exposed to β-HCH?

If the authors want to claim that β-HCH can increase the development of melanoma, they need to show that either β-HCH facilitates melanocyte transformation by oncogenes (such as BRAF, NRAS or KIT) or that β-HCH modulates the responses of melanocytes to UV, the predominant risk factor for melanoma. They could for example evaluate the effect of β-HCH on apoptosis or DNA repair induce by UV.

Figure 3a, b, c: are the changes in ROS, JC1 and mitotracker induced by β-HCH significant?

Figures 4 and 5: To evaluate the effect of β-HCH on mTOR, S6, AKT and AMPK phosphorylation, the authors need to show the corresponding blots for the total mTOR, S6, AKT and AMPK proteins and then quantify the ratio phosphorylated protein/ total protein for each protein. This is particularly important as Figure 5d shows a great variation in the total amount of mTOR and hence calculating the ratio pmTOR/mTOR may demonstrate that there is no activation of mTOR by phosphorylation despite what is claimed line 451.

Figure 4: considering the importance of the MAPK signaling pathway in melanocytes and melanoma, it seems essential to evaluate the effect of β-HCH on ERK phosphorylation.

Figure 5d: what is the effect of a-MSH +/- β-HCH on AKT phosphorylation upstream of S6 ? Can it explain the discrepancy between mTOR and S6 phosphorylation?

Figure 5e, f: the legends on the left side of the figures are missing.

Author Response

Cells

Editorial Office

                                                                          Rome, January 26th, 2024

Revised manuscript cells-2813929

Dear Editor,

please find enclosed the revised version of the manuscript (cells-2813929-R1) entitled “Persistent β-Hexachlorocyclohexane Exposure Impacts Cellular
Metabolism With a Specific Signature in the Human Normal Melanocytes:
Implication in Melanomagenesis” by F. Papaccio et al. Firstly, we would like to thank you and the reviewers for the positive and helpful comments about our paper. We have revised the manuscript accordingly to the requests and answered the comments point-by-point as follows:

Reviewers' comments:

Reviewer 2

The authors studied the effect of the organochlorine pesticide β-hexachlorocyclohexane (β-HCH) on melanocytes, skin fibroblasts, and keratinocytes in vitro. They showed that β-HCH increased keratinocytes and fibroblasts' proliferation whereas it inhibited melanocytes' proliferation. They further demonstrated that β-HCH selectively affected melanocyte intracellular metabolism, reducing ATP and inducing a general perturbation of redox equilibrium with a consequent reduction of melanogenesis. This was associated with a decrease in phosphorylation of several proteins involved in key signaling pathways. They concluded that exposure to β-HCH impaired melanocyte metabolism and could increase skin cancer risk.

The results are interesting to better understand the long-term effects of pesticides on skin cells. However, despite what is claimed in the title and the abstract, there is no evidence for an association of the impaired melanocyte metabolism induced by β-HCH and the increased in skin cancer risk. The authors clearly show that β-HCH inhibits melanocyte proliferation and inhibits the activation of several oncogenic pathways, which is not what you expect from an increased risk of melanocyte transformation on the contrary. From the data presented, in particular the reduction in melanin, one would expect β-HCH to induce depigmentation in vivo such as vitiligo. Is there any epidemiological evidence for increased vitiligo in population exposed to β-HCH?

- Thank you for the focused comment. In vitiligo, the white patches appearance is the consequence of selective melanocyte loss, rather than alteration of melanin synthesis or distribution. Many provoking factors have been described for melanocyte disappearance and the occurrence of the disease. Among these, genetic, biochemical, immunological and environmental factors converge in autoimmune destruction of melanocytes. However, epidemiological data about the role of exposure or contact with pesticides are scarce. In 1994, an epidemiological report, involving agricultural workers directly exposed to pesticides, excluded that several dermatological diseases, including vitiligo occur with different frequency compared to the control group. (doi: 10.1159/000246815). Further, very recently, (February 2024) the Vitiligo-linked Chemical Exposome Knowledgebase (ViCEKb) study using a systematic review method of the literature reported a list of 113 chemicals triggers of vitiligo that excluded β-HCH and/or similar compounds (Chivukula et al., 2024). By contrast, as presented in the introduction, elevated level of organochlorine pesticides correlate to melanoma increased risk (10.1002/cnr.1536).

If the authors want to claim that β-HCH can increase the development of melanoma, they need to show that either β-HCH facilitates melanocyte transformation by oncogenes (such as BRAF, NRAS or KIT) or that β-HCH modulates the responses of melanocytes to UV, the predominant risk factor for melanoma. They could for example evaluate the effect of β-HCH on apoptosis or DNA repair induce by UV.

-The possible role of β-HCH in the onset of melanoma probably deserves further studies that take into account even longer periods than those used in this work. As regards melanocytes, the only skin cells that showed suffering in the presence of β-HCH in our study, are not subject to apoptosis after continuous treatment with this pesticide. The apoptosis assay was presented in Fig.1. Regarding the DNA damage, we collected some preliminary data not included in the manuscript. Surprisingly we documented a marked reduction of pγ2AX (but also of the total amount of this DNA damage marker) as well as of other protein such as 53BP1. However, additional experiments are required to define a possible pro-tumorigenic role.

Figure 3a, b, c: are the changes in ROS, JC1 and mitotracker induced by β-HCH significant?

-As requested, we modified the histograms relative to the changes in ROS, JC-1, and mitotracker. By averaging results of all melanocyte cell lines we calculated statistical significance between -HCH-treated cells and untreated cells. Accordingly, Fig.3 a-c has been implemented.

Figures 4 and 5: To evaluate the effect of β-HCH on mTOR, S6, AKT and AMPK phosphorylation, the authors need to show the corresponding blots for the total mTOR, S6, AKT and AMPK proteins and then quantify the ratio phosphorylated protein/ total protein for each protein. This is particularly important as Figure 5d shows a great variation in the total amount of mTOR and hence calculating the ratio pmTOR/mTOR may demonstrate that there is no activation of mTOR by phosphorylation despite what is claimed line 451.

-As requested data regarding phosphorylated proteins have been associated to the images and densitometric analysis of total amount of the same protein. More, the ratio p-protein/total protein has been indicated by histograms. All the results have been discussed in the result and/or discussion sections.

Figure 4: considering the importance of the MAPK signaling pathway in melanocytes and melanoma, it seems essential to evaluate the effect of β-HCH on ERK phosphorylation.

-The level of phosphorylation of ERK has been added to fig.4a, as requested. It demonstrates activation of this signaling in presence of β-HCH.

Figure 5d: what is the effect of a-MSH +/- β-HCH on AKT phosphorylation upstream of S6? Can it explain the discrepancy between mTOR and S6 phosphorylation?

-The a-MSH activates mTOR, S6 as well as AMPK and AKT, however considering the ration Phospho/total protein significant differences between aMSH alone and aMSH plus b-HCH occurred only for AMPK and S6 indicating. Overall, energetic requirement for melanogenesis might be compromised.

Figure 5e, f: the legends on the left side of the figures are missing.

-We apologize. The legends have been added. 

Reviewer 3 Report

Comments and Suggestions for Authors

In a study by Papaccio et al, the Authors investigated β-Hexachlorocyclohexane exposure in normal human skin cells. They concluded that long-term exposure to persistent organic pollutants damages melanocyte metabolism in the function of the melanin content with a consequent reduction of melanogenesis indicating a potential augmented skin cancer risk.  

1. All figures must be reorganized and prepared once again as they are hardly to read.

2. Fig. 4 - selection of the proteins involved in the metabolism is not clearly stated and not justified as these proteins are involved in a number of cellular processes. More specific metabolic assays should be included.

3. The Authors state in the title "implication in melanomagenesis". While these results might have an implication in the development of melanoma, this has not been studied directly and should be removed from the title.

4. The results should be discussed in light of more recent papers, too.

Overall, the manuscript is interesting and the results are valid, however, their presentation must be improved.

Author Response

Cells

Editorial Office

                                                                          Rome, January 26th, 2024

In a study by Papaccio et al, the Authors investigated β-Hexachlorocyclohexane exposure in normal human skin cells. They concluded that long-term exposure to persistent organic pollutants damages melanocyte metabolism in the function of the melanin content with a consequent reduction of melanogenesis indicating a potential augmented skin cancer risk.  

  1. All figures must be reorganized and prepared once again as they are hardly to read.

-As suggested, we revised the figures layout. However, if some additional modification are essential for the Reviewer we will change further the images.

  1. 4 - selection of the proteins involved in the metabolism is not clearly stated and not justified as these proteins are involved in a number of cellular processes. More specific metabolic assays should be included.

-  In accordance with the reviewer’s comment, we improved western blot analyses on melanocyte cell lines to check more specifically modifications on metabolic pathways. The protein level of several glycolytic enzymes, such as Hexokinase 1 and 2, Pyruvate kinase (PKM1/2) and of Platelet-type phosphofructokinase were lessened by b-HCH exposure. By contrast, total OXOPHOS mitochondrial complexes investigated by western blot resulted unmodified.

  1. The Authors state in the title "implication in melanomagenesis". While these results might have an implication in the development of melanoma, this has not been studied directly and should be removed from the title.

We agree with the reviewer and reconsidering the overall data presented we modified the title deleting "implication in melanomagenesis".

  1. The results should be discussed in light of more recent papers, too.

- We thank the Reviewer for this suggestion. We checked again the most recent literature. Accordingly, we enriched the introduction and the discussion, including more recent reports about the effects of -HCH. The added parts have been underlined in the text.

Round 2

Reviewer 2 Report

Comments and Suggestions for Authors

Suitable for publication

Reviewer 3 Report

Comments and Suggestions for Authors

Comments have been adequately addressed